# k-dependent modulation of intrinsic spin-orbit interaction in MoSe$_2$ induced by proximity to amorphous Pb

Fatima Alarab [1], Ján Minár [2] ✉, Procopios Constantinou [1], Dhani Nafday[3], Aki Pulkkinen [2], Thorsten Schmitt [1], Xiaoqiang Wang[1] & Vladimir N. Strocov [1] ✉

Tuning the strength of spin-orbit interaction (SOI) is pivotal for developing next-generation spintronic and quantum devices. Proximity-induced SOI is a promising route toward this goal, but its experimental characterization with resolution in **k**-space using ARPES remains challenging. We advance previous ARPES investigations of proximity-induced SOI in graphene-based systems to transition-metal dichalcogenides (MoSe$_2$) in proximity to an amorphous overlayer of high-$Z$ metal (Pb) whose disordered nature suppresses **k**-space mismatch at the interface. The use of soft-X-ray ARPES is instrumental for accessing MoSe$_2$ beneath the Pb layer. We introduce an approach to interpret the experimental data based on the identification of local SOI-derived band gaps—SOI hotspots—where the intrinsic SOI contribution, arising from the SOI field transfer from the overlayer to the host, is isolated from competing effects such as scalar (non-SOI) hybridization, interlayer interactions and Rashba-type splitting. We find that the proximity to Pb strongly enhances the intrinsic SOI as manifested by **k**-dependent increase of the band splitting in the SOI hotspots by up to several tens of meV. Tunability of this effect via Pb coverage provides versatile means for tailoring SOI to specific spintronic and quantum applications.

Spintronics is based on novel operational principles that utilise the electron spin degree of freedom to process information, offering a greater functional diversity with respect to conventional electronics[1–3]. Spintronic devices are also expected to be the leading contenders for the next-generation devices, which will boast reduced power consumption, increased memory density, and processing capabilities. Typically, in such spintronic devices, spin polarisation is controlled either by magnetic layers or via spin-orbit (SO) interaction (SOI)[4].

SOI is known to play a key role in a variety of novel functionalities that distinguish materials used for spintronics. Depending strongly on the SOI strength, these functionalities include spin-orbit torques[3], the anomalous Hall effect[5], anisotropic magnetoresistance[6], and spin

relaxation[7] in magnetic materials. They also encompass interconversion between nonequilibrium charge and spin currents[8,9] as well as the direct and inverse Rashba–Edelstein effects in non-magnetic materials[10,11]. Strong SOI is a key requirement for forming topologically protected electronic states, including two-dimensional (2D) surface states in topological insulators[12,13] and three-dimensional (3D) states such as nodal lines and chains in Dirac and Weyl semimetals[14,15]. Pivotal for detailed investigation of such states has been angle-resolved photoelectron spectroscopy (ARPES) as a unique experimental technique resolving electron states in electron momentum (**k**). Beyond spintronics, tunable SOI is vital for engineering exotic quantum states such as Majorana fermions, envisioned for qubits where

[1]Swiss Light Source, Paul Scherrer Institute, Villigen-PSI, Switzerland. [2]New Technologies Research Centre, University of West Bohemia, Plzeň, Czechia. [3]Asia Pacific Center for Theoretical Physics, Pohang, Gyeongbuk, South Korea. ✉e-mail: jminar@ntc.zcu.cz; vladimir.strocov@psi.ch

superconductors are interfaced with strong-SOI semiconductors[16,17]. Here, SOI provides topological protection against decoherence from low-energy excitations[18]. Therefore, identifying practical and controllable ways to tune SOI strength is critical for advancing functional materials for next-generation spintronic and quantum devices.

The SOI strength can be increased by interfacing materials with high-$Z$ atoms. This so-called proximity effect manifests as an enhancement of the SO splitting ($\Delta_{SO}$) in the host band structure. We can differentiate two contributions to the proximity effect[19–22]: (1) Rashba-type SOI, which introduces energy splitting proportional to the Rashba parameter ($\alpha_R$), which, in the heuristic form, is written as $\alpha_R \sim \lambda_0 |\langle \psi_0 | \frac{\partial V}{\partial z} | \psi_0 \rangle|$. Here, $\lambda_0$ is the SOI constant of the host atoms, $\psi_0$ the host wavefunction, and $\frac{\partial V}{\partial z}$ the potential gradient. Proximity to the external atoms can render $\alpha_R$ non-zero in two different ways, either by a structural distortion causing asymmetry of $\psi_0$ (often referred to as structure-induced Rashba effect) or by imposing an external electric field on top of the host $\frac{\partial V}{\partial z}$ (field-induced Rashba effect). In either case, the Rashba-type SOI can be identified based on the characteristic doublet of band dispersions with opposite spin, which are split in $\mathbf{k}$-space[23]; (2) the so-called intrinsic SOI, where $\psi_0$ hybridizes with that of the high-$Z$ external atom ($\psi_1$), providing a strong SOI field. In this case, energy splitting is proportional to the proximity-enhanced SOI strength ($\lambda_{eff}$), which, in a simplified form of the second-order perturbation theory, is written as $\lambda_{eff} \sim \lambda_0 + \frac{|\langle \psi_0 | H_1^{SOI} | \psi_1 \rangle|^2}{E_0 - E_1}$, where $H_1^{SOI}$ and $E_1$ are the spin-orbit Hamiltonian and energy level of the external atoms, respectively. While several studies have reported proximity-induced SOI, clear experimental identification of the intrinsic contribution remains rare. However, under favourable hybridization conditions and high-$Z$ external atoms, introducing a strong SOI field, the intrinsic contribution should dominate the proximity effects. An insightful theoretical discussion of the Rashba-type vs intrinsic SOI, tailored to graphene-based materials in proximity to transition-metal dichalcogenides (TMDCs), can be found in refs. 19–22.

Most experimental studies of proximity effects focused on graphene-related systems. For example, numerous ARPES studies of graphene grown on various $5d$ metal substrates such as Au, Ir, and Pt[24–29] have detected tangible structure-induced Rashba splitting coexisting with band gaps opening due to scalar (non-SOI) hybridization of graphene with substrate states. However, the proximity-induced intrinsic SOI could not be convincingly disentangled from Rashba-type effects. An alternative strategy to manipulate SOI in graphene is its interfacing to TMDCs. Magnetotransport measurements[30] and scanning tunneling spectroscopy[31] identified the Rashba and the intrinsic valley-Zeeman terms in the proximity-induced SOI in such heterostructures, with magnitudes of the order of 10 meV and 2 meV, respectively. Such effects have also been observed in heterojunctions MoSe$_2$/EuS[32], graphene on WS$_2$[33] and other TMDCs[34], and bilayer graphene on MoS$_2$[35] and WSe$_2$[36], with strong impact on spin-relaxation anisotropy[33,37]. Inversion symmetry can also be broken by applying an out-of-plane electric field, but even at several V/nm, the resulting spin splitting remains at the few-meV level[10,21]. In any case, the gapless nature of pristine graphene questions its utility for transistor applications.

An alternative platform to manipulate the SOI effects can be the diverse family of TMDCs, composed of weakly van der Waals (vdW) bonded atomic layers[38,39]. Various band structure patterns, including semiconductor band gaps, make these materials promising candidates for novel microelectronic and spintronic devices. One such TMDC is MoSe$_2$, an indirect band gap semiconductor crystallising in its thermodynamically stable 2H-phase shown in Fig. 1a, b. Its trigonal single-layer structure denies inversion symmetry, allowing intrinsic SOI to cause energy- and $\mathbf{k}$-dependent splitting of the valence bands of the order of ~200 meV. Owing to this distinct splitting, MoSe$_2$ is an ideal host for ARPES experiments on proximity-induced modulation of SOI. Theoretically, Rashba-type SOI spin splitting has been predicted for

PtSe$_2$/MoSe$_2$ heterostructures[40]. Magnetic proximity effects in graphene-based materials and TMDCs[32,40–43] are beyond the scope of this work.

Here, we extend previous ARPES studies of proximity-induced SOI from graphene-based materials as receivers of the SOI field to a new class of materials, TMDCs (specifically, 2H-MoSe$_2$). In contrast to previous epitaxial systems, the high-$Z$ overlayer (Pb) is amorphous, which avoids $\mathbf{k}$-space mismatch at the interface, otherwise limiting proximity effects. Methodologically, we introduce an approach based on the identification of isolated SOI-derived band gaps—SOI hotspots—that allows separation of the intrinsic SOI contribution, arising from the SOI field transfer, from competing effects such as scalar hybridization and Rashba-type splitting. Applying this framework to the MoSe$_2$/Pb interface, we observe proximity-induced enhancement of intrinsic SOI by several tens of meV tunable via the Pb thickness. Exceeding the effects reported for graphene-based systems, this enhancement reflects a genuine intrinsic proximity mechanism driven by hybridization between Mo $d$-orbitals and Pb $sp$-orbitals, which imposes a strong SOI field on MoSe$_2$. Our experiments use soft-X-ray ARPES at photon energies ($h\nu$) around 800 eV, where large photoelectron inelastic mean free path (IMFP) enables access to MoSe$_2$ beneath the Pb layer and also sharpens the intrinsic out-of-plane momentum resolution ($\Delta k_z$) for precise navigation in 3D $\mathbf{k}$-space[44–47].

## Results and discussion

### SO splitting hotspots in the valence band of bare MoSe$_2$

Similarly to other TMDCs[48–51], the valence band (VB) of MoSe$_2$ is formed by the metal states (Mo $4d$) hybridized with the chalcogen ones (Se $4p$). The electronic structure of MoSe$_2$ critically depends on the number of layers upon their pile-up from one monolayer to the bulk[51]. The corresponding evolution of the electronic structure is driven by a modification in the hybridization between the out-of-plane Se $3p_z$ orbitals, in turn interacting with the Se $4p_{xy}$ in-plane orbitals and with the Mo $4d$ ones inside the layers. This hybridization is responsible for the interlayer interaction in bulk MoSe$_2$ and quantum confinement in its heterostructures. Furthermore, in systems with an odd number of MLs lacking the inversion centre, this hybridization modulates the Rashba-type SO splitting at the Γ-point[23,51]. A particular focus of previous works has been the local VB maximum in the K-point of the monolayer through bulk MoSe$_2$, see the corresponding 2D and 3D Brillouin zones (BZs) in Fig. 1c. It was shown that the band splitting ($\Delta_{bs}$) at this point arises mostly from the interlayer interaction with a smaller contribution from the intrinsic SOI and, in the monolayer limit, from the strong SOI caused by the absence of the inversion centre[51–53].

Our DFT calculations for bulk MoSe$_2$ without and with SOI are shown in Fig. 2a, b for the in-plane directions and in (c,d) for the out-of-plane ones. The ΓA direction in (c,d) is unfolded to the $\Gamma_0 A \Gamma_1$ one, whose length is the double out-of-plane size of the 3D-BZ. This double-BZ periodicity reflects the actual periodicity of the ARPES response of 2H-MoSe$_2$ related to a non-symmorphic space group of its crystal structure[54,55]. Hereafter, energy scale of all theoretical and experimental data is set relative to the global VB maximum (VBM) in the $\Gamma_1$-point.

The SOI effects in the band structure of MoSe$_2$ can be distinguished in Fig. 2 by comparing (a) with (b). At the K-point maximum (green circle), $\Delta_{bs}$ increases only slightly when SOI is added. At this point, it is therefore controlled mostly by the interlayer interaction with only a weak contribution of SOI. The H-point, which differs from the K only in $k_z$, is completely different, however. The bands near $E_F$ at this point (red circle) split only when SOI is activated. Therefore, in contrast to the sister K-point, $\Delta_{bs}$ at this point results solely from strong SOI. This makes the H-point the ideal hotspot for analysis of SOI and its modulation driven by the proximity effect. Another band structure point where the SOI effects are critical is the band manifold at binding energy ($E_B$) around −1 eV at the Γ-point (blue circle).

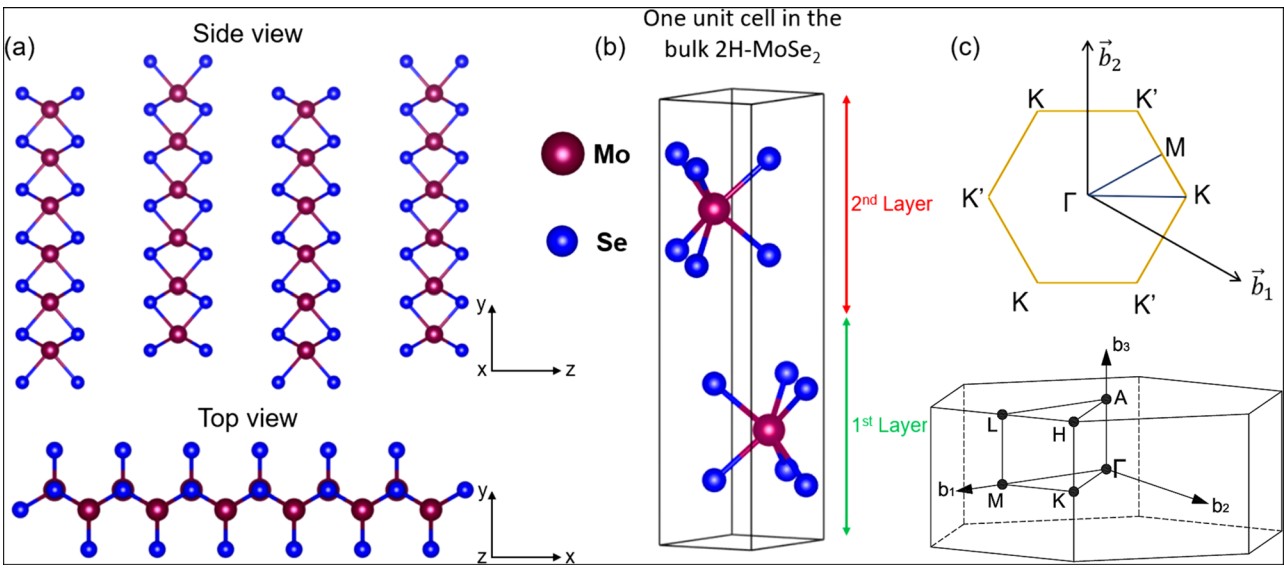

**Fig. 1 | Crystallography of 2H-MoSe₂. a** Side- and top-view of MoSe₂ crystal structure in its prismatic 2H-phase. **b** One unit cell in the bulk 2H-MoSe₂. **c** Corresponding 2D (top) and 3D (bottom) BZs of MoSe₂ showing the high-symmetry points.

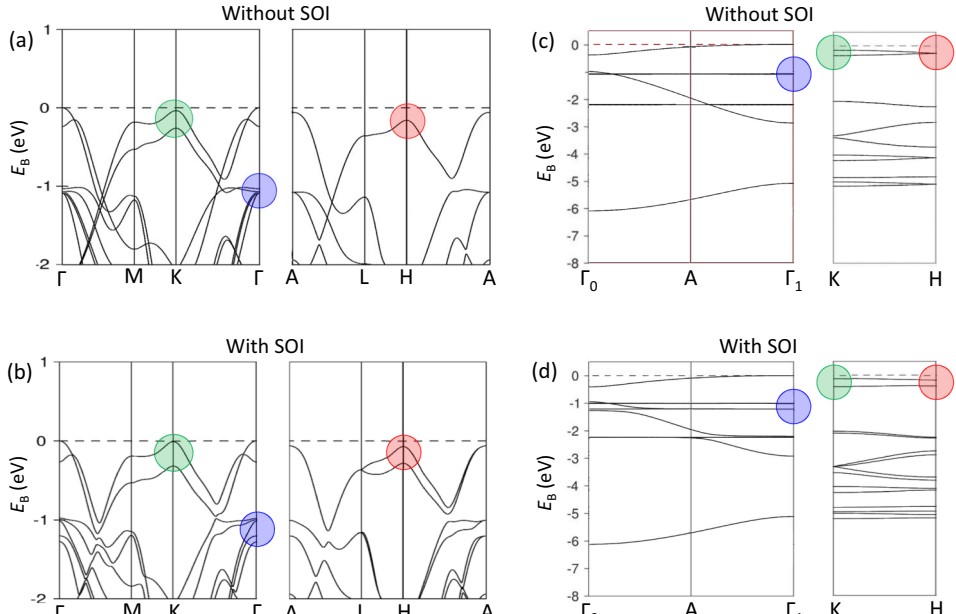

**Fig. 2 | DFT band structure of bulk MoSe₂. a**, **b** In-plane symmetry directions, and **c**, **d** out-of-plane ones without and with SOI (the ΓA direction unfolded to the double out-of-plane size of the 3D-BZ). The energy scale hereinafter is relative to the VBM in the Γ₁-point. The band splitting in the K-point (green circle) is largely

due to the interlayer interaction, and those in the H- and Γ₁-points (red and blue circles, respectively) are the two hotspots where the splitting is predominated by SOI.

Whereas without SOI this manifold is almost degenerate in the Γ-point, it spills over an energy interval of ~250 meV when SOI is on. The unfolded band structures along $\Gamma_0 A \Gamma_1$, in (c) and (d), show that the point most relevant for our analysis is $\Gamma_1$, where the SOI-split doublet is maximally separated in energy from other bands. This is therefore the second hotspot of SOI. Yet another hot spot might in principle be found at the K-point at $E_F$ around −3.5 eV, but the corresponding $\Delta_{bs}$ will be hidden in the spectral peak broadening which increases away from $E_F$ because of decreasing hole lifetime. A key property of the SOI hotspots, as we will see below, is that the band gap magnitude is insensitive to scalar (non-SOI) band hybridization and Rashba-type spin splitting, making them ideal for isolating the intrinsic proximity effects.

## Experimental valence band of bare MoSe₂

To identify the SOI hotspots in the experimental ARPES data, we will first establish the experimental 3D band structure of bare MoSe₂. Figure 3a shows the $(k_x, k_z)$ out-of-plane iso-$E_B$ maps of ARPES spectral intensity taken at $E_B = 0$ and −0.65 eV under variation of $hv$. A dispersive periodic pattern along $k_z$ through the consecutive Γ-points manifest 3D electronic states in this **k**-space region, with the observed $k_z$ periodicity of the double out-of-plane size of the 3D-BZ reflecting the non-symmorphic space group of 2H-MoSe₂. The nearly non-dispersive lines through the K-points manifest nearly 2D states consistent with the calculations in Fig. 2c, d. The conversion from $hv$ to $k_z$ used the free-electron approximation for the final states (corrected for the photon momentum) with an empirical inner potential $V_0$ of 14 eV.

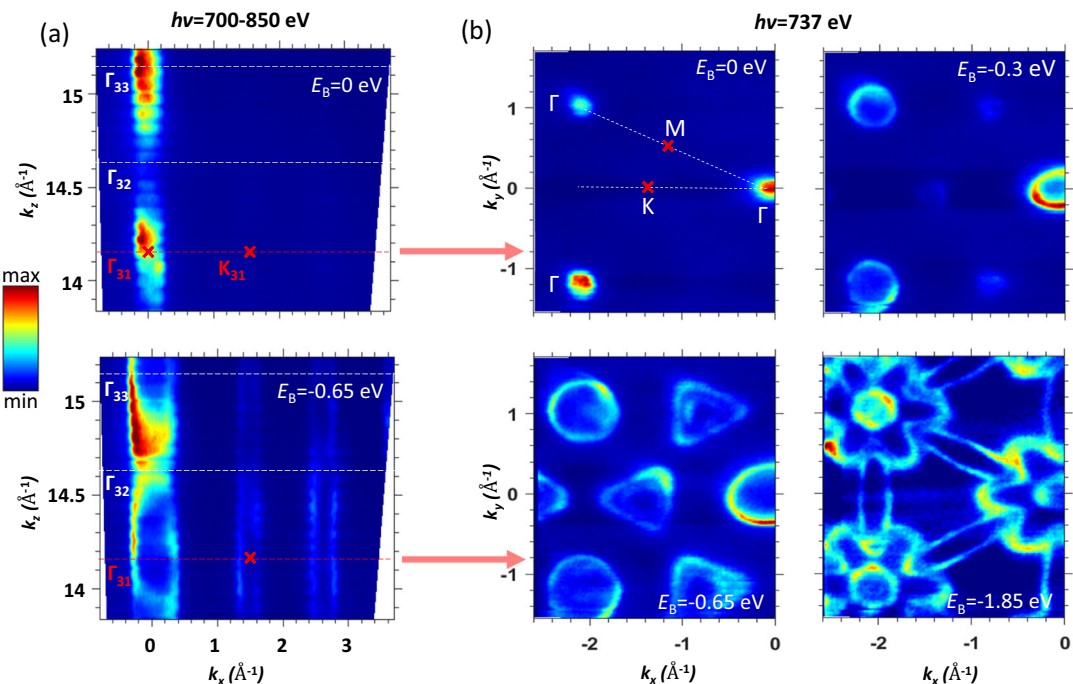

**Fig. 3 | Experimental iso-$E_B$ maps for bare MoSe$_2$. a** Out-of-plane maps of ARPES intensity in the $h\nu$ range 700–850 eV, taken for $k_x$ oriented along the $\Gamma_1$K direction of the 3D-BZ (Fig. 1c) at $E_B = 0$ and −0.65 eV. **b** The in-plane maps with the same $k_x$ orientation at $E_b = 0$, −0.3, −0.65 and −1.85 eV. These maps were taken at $h\nu = 737$ eV, corresponding to the $\Gamma_1$KM plane of the 3D-BZ. The hexagonal symmetry and the trigonal warping around the K-points are seen.

Figure 3b shows an additional sequence of $(k_x,k_z)$ in-plane maps of the ARPES intensity for a few $E_B$ values, which were measured at $h\nu = 737$ eV corresponding to the $\Gamma_1$ point in the unfolded band structure, Fig. 2c. The map for $E_B = −0.65$ eV, in particular, shows up a hexagonal pattern of circles around the $\Gamma_1$-points and closed trigonal contours around the K-points, with the latter split because of the aforementioned interlayer splitting. This effect is in agreement with our calculations in Fig. 2b as well as with the previous theoretical and experimental data[35,51,56,57].

The experimental out-of-plane dispersion of the valence states is represented in Fig. 4a, which shows the ARPES intensity maps as a function of $E_B$ and $k_z$ measured at $k_x = 0$ Å$^{-1}$ (along $\Gamma$A) and 1.6 Å$^{-1}$ (along KH) in the $h\nu$ range from 700 eV to 850 eV. Along $\Gamma$A, the two bands with the strongest $k_z$-dispersions are formed by the bonding Se $4p_z$ and antibonding Se $4p_z^*$ out-of-plane orbitals (centered at $E_B \sim −6.3$ eV and −2.5 eV, respectively) which overlap across the vdW gap. The dispersing band near the VBM is formed by the Mo $4d$ orbitals located inside the atomic layers, and its $k_z$-dispersion manifests an interlayer cross-talk of these orbitals mediated by their hybridization with the Se $4p_z^*$ ones[48]. For the $\Gamma$A direction, the experimental dispersions run the indicated sequence through the $\Gamma$-points (the even indices correspond to $\Gamma_0$ and odd to $\Gamma_1$) separated by the A-ones. The observed $k_z$ periodicity, following the double-BZ periodicity, agrees with the calculated band dispersions unfolded to $\Gamma_0$A$\Gamma_1$ in Fig. 2c, d. For the KH direction, the experimental dispersions are more flat, and the double-BZ periodicity is less clear. The experimental out-of-plane ARPES intensity as a function of $E_B$ and $k_x$ along the $\Gamma$K direction is represented in Fig. 4b. The left one was measured at $h\nu = 737$ eV (the same as used for the in-plane iso-$E_B$ maps in Fig. 3b), which brings to $k_z$ at the $\Gamma_1$-point. Due to small variations of $k_z$ with $k_{xy}$ in the soft-X-ray range, this map corresponds almost exactly to the $\Gamma_1$K direction of the 3D-BZ. The right map was measured at 715 eV, bringing $k_z$ to the A-point, and corresponds to AH.

We will now focus on $\Delta_{bs}$ at the $\Gamma_1$- and H-hotspots, which are central for our analysis of SOI. Figure 4c, d shows zoom-ins of the ARPES data from (b) around these hotspots. The experimental bands at the $\Gamma_1$- and H-hotspots are clearly split, which is also evident in the corresponding Energy-Distribution Curves (EDCs) plotted next to the zoom-ins. The $\Delta_{bs}$ values were determined from the negative second derivative of these EDCs ($−d^2I/dE^2 > 0$, with the unphysical negative values set to zero) shown to the right. These values, in our case reflecting almost pure $\Delta_{SO}$, were found as 221 ± 2 meV in the $\Gamma_1$-hotspot and 210 ± 2 meV in the H-one. They are consistent with our DFT calculations with SOI in Fig. 2b. We note that the H-point differs from the K-one, where $\Delta_{bs}$ is driven primarily by the interlayer interaction, only by its $k_z$. The small intrinsic $\Delta k_z$ achieved in the soft-X-ray energy range is quintessential to distinguish these two points in the experimental spectra[45–47]. Indeed, according to the TPP-2M formula[58,59], the IMFP in our kinetic-energy range is ~15 Å. This gives $\Delta k_z \sim 0.07$ Å$^{-1}$, which is sufficiently sharp relative to the $k_z$ separation of the K- and H-points of 0.23 Å$^{-1}$. We notice, however, that while our calculations predict $\Delta_{bs}$ to vary from ~290 to 200 meV between the K- and H-points, the experiment finds it stay at 210 meV, essentially constant within the experimental accuracy.

**Electronic structure evolution with Pb thickness**

To study the proximity-induced SOI in MoSe$_2$, we deposited Pb in-situ on its cleaved crystalline surface, and tracked the evolution of $\Delta_{bs}$ at the two SO-hotspots versus the overlayer thickness. As Pb does not wet the MoSe$_2$ surface, the depositions from a thermal evaporator were carried out at the sample temperature 14 K to ensure formation of a continuous amorphous overlayer, essential for our study. The cumulative depositions were each followed by X-ray photoemission spectroscopy (XPS) and ARPES measurements. The XPS spectra of the Pb $4f$ and Mo $3d$ core-level peaks, Fig. 5a, have shown a monotonous increase of their intensity ratio upon the Pb deposition corresponding

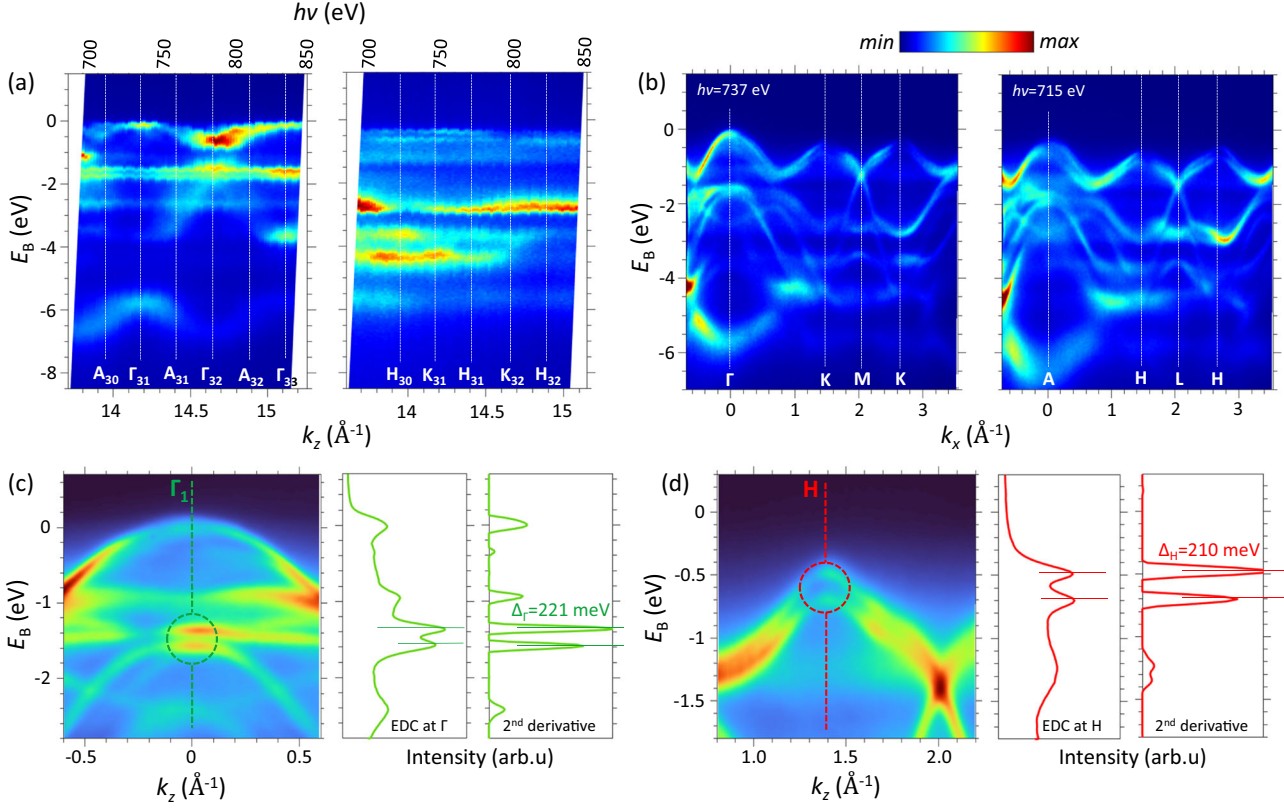

**Fig. 4 | Experimental band structure and intrinsic SO splitting for bare MoSe₂.** **a** Out-of-plane ARPES intensity map as a function $k_z$ in the $h\nu$ range 700–850 eV along the ΓΑ and ΚΗ directions. **b** In-plane ARPES intensity images measured at $h\nu = 737$ eV (left) and 715 eV (right), where **k** runs along the $\Gamma_1$K and AH directions of the 3D-BZ. **c**, **d** Zoom-ins of **b** near the $\Gamma_1$- and H-hotspots (marked) with the corresponding EDC and $-d^2I/dE^2 > 0$ plots in these points. The horizontal dashes on the $-d^2I/dE^2 > 0$ plots mark the corresponding $\Delta_{bs}$ values, in this case identified with $\Delta_{SO}$.

to a linear increase of the overlayer thickness with the deposition time without formation of islands. Here, a thickness of 2.86 Å corresponds to one effective monolayer (ML) of Pb[60]. Importantly, the Mo 3$d$ and Se 3$d$ peaks have not shown any noticeable lineshape changes upon the deposition, confirming the absence of chemical reactions between the Pb and MoSe₂ or formation of new chemical environments of the interfacial atoms. This observation also excludes any intercalation of Pb atoms into MoSe₂ as well as layer-breaking reactions[61].

ARPES data in the VB region of MoSe₂ under the Pb overlayer is represented in Fig. 5b as measured for an overlayer thickness of 0.25 nm (corresponding to ~1 ML). The full dataset measured along the ΓΚΜΚ and AHLH lines through the whole series of Pb depositions is represented in the Supporting Information (SI) in Fig. SI1. We notice a significant incoherent background coming from the amorphous Pb layer (Fig. SI2) and increased broadening of the spectral peaks due to partial disorder induced into MoSe₂ by the Pb atoms. Figure 5c shows the data from (b) overlaid with the bands of bare MoSe₂ from Fig. 4b in the curvature representation[62]. The observed energy shifts are **k**-dependent and are of the order of 100 meV and less. Their main cause is that the electron density introduced by Pb atoms mediates the scalar orbital hybridization[63] in MoSe₂; another effect, discussed below, is the SOI field. Besides these effects, the band structure of MoSe₂ stays quite resilient under the Pb overlayer, without formation of additional coherent electronic states. Also notable in the spectra is the Fermi edge from the Pb overlayer, best seen in angle-integrated spectra in Fig. SI1. Owing to the low density of Pb states in this energy region, the Fermi edge clearly identifies only at higher Pb coverages. It is located at ~400 meV above the VBM, varying with the Pb thickness within

±50 meV. This variation is less expected because most metal-TMDC interfaces show pinning of the Fermi level to the VBM, resulting from formation of intermetallic compounds at the interface[64,65]. Aligned with the absence of the core-level changes and resilience of the buried MoSe₂ band structure, this behaviour is attributed to low chemical activity of Pb on MoSe₂.

In order to identify the SOI modulation due to the proximity effects, we will now turn to the evolution of the band structure in the $\Gamma_1$- and H-hotspots upon increase of the Pb thickness. Following the in-plane ARPES data around these hotspots for bare MoSe₂ in Fig. 4e, f, the zoom-ins in Fig. 6a–d show the corresponding EDC and $-d^2I/dE^2 > 0$ data for MoSe₂ covered with 0.25 and 0.5 nm of Pb, equivalent to ~1 and 2 MLs, respectively (the full dataset compiling all Pb coverages is presented in Fig. SI3). The $-d^2I/dE^2$ peaks signal a significant increase of $\Delta_{bs}$ in the $\Gamma_1$-hotspot from 221 meV in bare MoSe₂ to 231 ± 2 and 252 ± 4 meV for the 0.25- and 0.5-nm thickness of the Pb overlayer, and in the H-hotspot from 210 meV to 250 ± 2 and 270 ± 2 meV.

Importantly, the $\Delta_{bs}$ modulations in the SOI hotspots are unaffected by scalar hybridization of the Pb states with MoSe₂, which contribute to the marginal **k**-dependent energy shifts[63] noticed in Fig. 5c. This is clear from the fact that the electron wavefunctions above and below the SOI-induced band gaps have the same spatial (spin-independent) parts and should therefore exhibit almost the same energy shifts upon the scalar hybridization, leaving $\Delta_{bs}$ the difference unchanged. The observed increase of $\Delta_{bs}$ is significantly stronger than for graphene also interfaced to high-$Z$ metals[24–28] or TMDCs[33–35].

We interpret the observed increase of $\Delta_{bs}$ in the SOI hotspots as due to the intrinsic proximity effect caused by the transfer of the

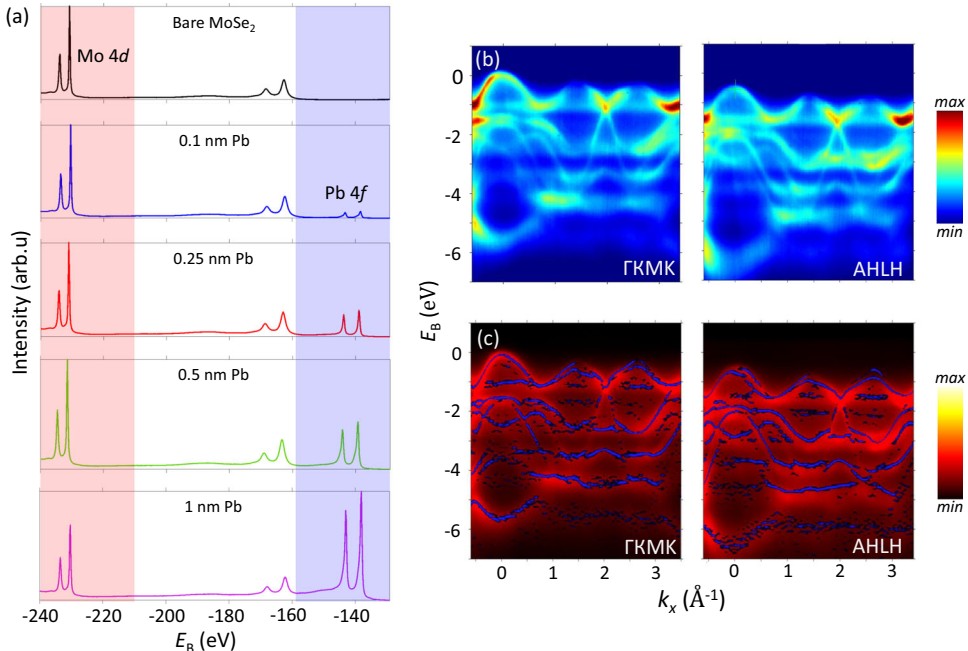

**Fig. 5 | Evolution of the core-level and VB spectra upon deposition of the amorphous Pb overlayer. a** XPS overview spectra at $h\nu = 1000$ eV showing the Mo $3d$ and Pb $4f$ core levels depending on the cumulative Pb thickness. **b** In-plane ARPES images along the $\Gamma_1 K$ and AH directions of the 3D-BZ for a Pb thickness of 0.25 nm, and **c** these images in red colorscale overlaid with the bands of bare MoSe$_2$ (blue). Besides an increase of the incoherent background, broadening of the spectral peaks and their marginal **k**-dependent energy shifts, the Pb overlayer does not much affect the MoSe$_2$ electronic states.

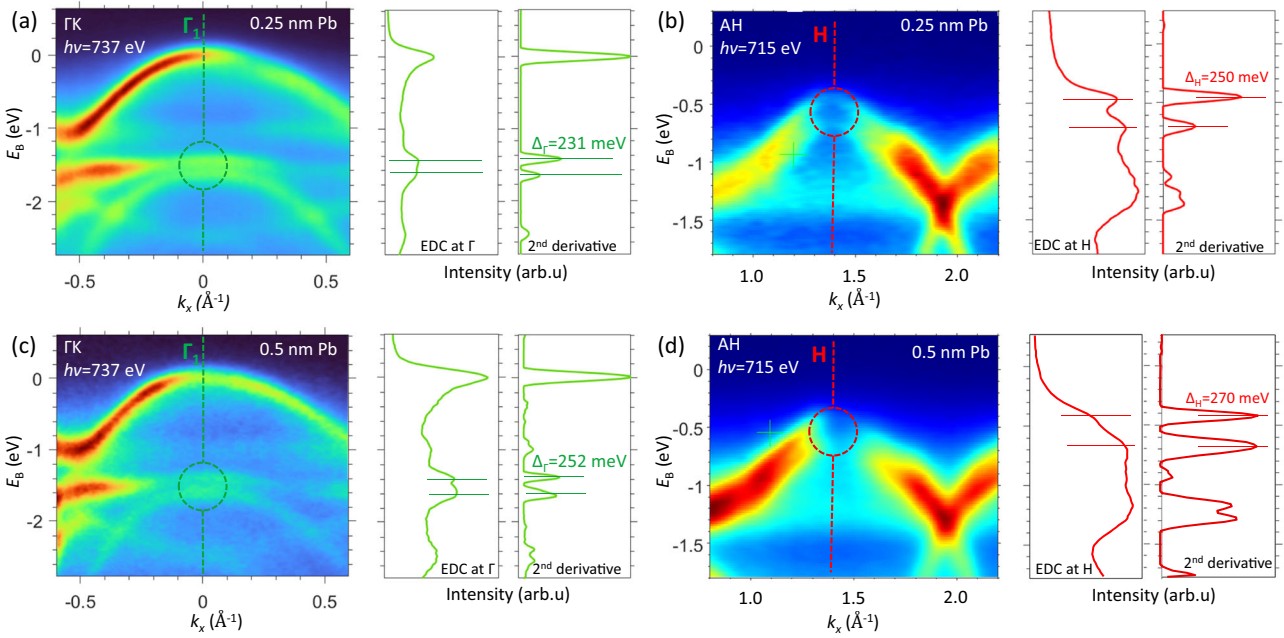

**Fig. 6 | SOI modulation at the Pb/MoSe$_2$ interface.** ARPES data for the $\Gamma_1$- and H-hotspots (**a**–**d** respectively) for the 0.25- and 0.5-nm thickness (**a**–**d**) of the Pb overlayer, represented in the same way as the data on bare MoSe$_2$ in Fig. 4c, d. The increase of $\Delta_{bs}$ upon the deposition of Pb identifies the increase of SOI in MoSe$_2$ caused by the proximity effect.

strong SOI field from Pb to MoSe$_2$. Its mechanism is the hybridization of the Pb wavefunctions with the Mo-derived ones across the Pb/MoSe$_2$ interface, mediated by the Se-derived wavefunctions in the top layer of MoSe$_2$. Below, we ascertain this interpretation based on assessment of alternative scenarios.

The whole body of our experimental data on $\Delta_{bs}$ as a function of the Pb thickness is compiled in Fig. 7. It runs over the $\Gamma_{31}$-, H$_{30}$- and H$_{31}$-hotspots from the experimental $k_z$ dependence in Fig. 4a, b for the Pb-layer thickness from 0 to ~2.5 nm (also included here is the $\Gamma_{32}$-point, measured at $h\nu = 778$ eV and corresponding to $\Gamma_0$ in Fig. 2d, although

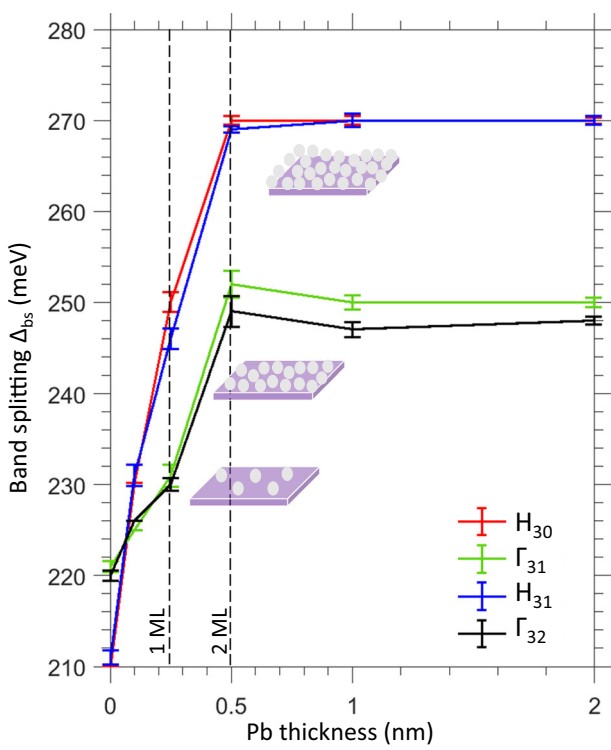

**Fig. 7 | Pb-thickness dependence of the SO band splitting at the Pb/MoSe₂ interface.** Evolution of $\Delta_{bs}$ as a function of Pb-overlayer thickness through the indicated equivalent $\Gamma_1$- and H-hotspots. The confidence interval widths for the $\Delta_{bs}$ values are indicated by vertical bars. $\Delta_{bs}$ saturates at a Pb thickness of 0.5 nm (about two full MLs).

$\Delta_{bs}$ in this point is affected by hybridization with the Se $4p_z{}^*$ band). We see that as the Pb thickness piles up, the experimental $\Delta_{bs}$ gradually increases and saturates at about two full MLs. In the H-hotspots $\Delta_{bs}$ saturates at ~270 meV (increasing by ~60 meV compared to pristine MoSe₂), and in the $\Gamma_1$-hotspot at ~250 meV (increasing by ~30 meV).

Interestingly, $\Delta_{bs}$ continues to increase beyond 1 ML coverage, despite the short-range nature of SOI. This behavior can be attributed to Pb atoms adsorbing on already covered regions and leaving uncovered areas accessible to atoms from the continuing Pb. Also, we cannot rule out an amplification of the interfacial SOI field from interactions among Pb atoms within the deposited film. The slight reduction of $\Delta_{bs}$ for the $\Gamma_{31}$ and $\Gamma_{32}$ points at the Pb thickness around 1 nm, though within the confidence intervals, could be attributed to a complex band hybridization pattern near these **k**-points. Evidently, the key parameter to control the proximity-induced $\Delta_{SO}$ is the Pb coverage, mostly in the sub-ML range.

## Assessment of alternative scenarios

We have also analysed alternative scenarios for the observed modulation of $\Delta_{bs}$ in the SOI hotspots. The simplest possibility might be a chemical reaction between Pb and MoSe₂. However, our XPS spectra show no evidence for new chemical environments formed at the interface, and the ARPES data reveal no formation of new electron states. This is consistent with the absence of the Fermi-level pinning to the VBM, indicative of the formation of intermetallic compounds[64,65]. These observations allow us to rule out chemical-reaction origins of the $\Delta_{bs}$ modulation.

Next, we considered whether the $\Delta_{bs}$ modulation could arise from Rashba-type SOI. Critically, the experimental bandstructure in Fig. 6 shows no sign of the characteristic **k**-space splitting of the bands in the SOI hotspots. Given the large increase of $\Delta_{bs}$, such splitting would be also large compared to the experimental **k**-resolution—yet not

observed. As additional arguments to rule out the Rashba-type SOI, we will now analyze its structure-induced and field-induced mechanisms.

Two considerations argue against the structure-induced Rashba-type SOI. First, it would typically require a well-defined interfacial asymmetry in the out-of-plane direction, as found in crystalline heterostructures. In contrast, the Pb overlayer in our system is amorphous and lacks long-range structural order, making such asymmetry unlikely. The second argument aligns with the insensitivity of the band gap in the SOI hotspots to the scalar hybridization, discussed above. Owing to the nearly identical spatial parts of the wavefunctions at the edges of the SOI-induced band gaps, the Rashba effect would affect them equally, leaving $\Delta_{bs}$ at most unchanged, in contrast to our experiment.

To assess the role of field-induced Rashba SOI, we performed band structure calculations for MoSe₂ under a static out-of-plane electric field up to 1 V/nm, an upper-bound estimate for the field potentially induced by the Pb overlayer. Our theoretical results in Fig. SI4-1 show that the field lifts the spin degeneracy of the bands, which we classify by their dominant out-of-plane spin component $S_z$. In this context, $\Delta_{bs}$ is defined as the energy separation between the centers of the spin-split doublets, marked by green dashes at the K- and H-points. Fig. SI4-2 shows that $\Delta_{bs}$ at the H-hotspot stays constant within a few µeV over the entire field range. This behavior contrasts to our experimental increase of ~60 meV upon Pb deposition. A similar lack of field effect is found at the $\Gamma_1$-point, again contradicting the ~30 meV increase observed in the experiment. These findings allow us to rule out any significant contribution from field-induced Rashba SOI to the experimental $\Delta_{bs}$ modulation. We note in passing that direct verification of the field effect in ARPES experiments[66] presently stays challenging because of overwhelming technical difficulties such as gate leakage currents.

Lastly, the clearest demonstration of the role of strong SOI field in the observed proximity effect has come from a control experiment in which we replaced the Pb deposition by Al, an element having comparable electronic structure but much lower atomic SOI. The resulting ARPES bandstructure is presented in Fig. SI5. Larger chemical activity of Al compared to Pb is evidenced in larger incoherent spectral background. Strikingly, no changes of $\Delta_{bs}$ in either SOI hotspot were detected upon the Al deposition. The whole body of our results, taken together and critically compared against alternative scenarios, confirms that the observed $\Delta_{bs}$ modulation is a genuine intrinsic proximity effect, arising from the transfer of a strong SOI field from Pb into MoSe₂.

In summary, we have demonstrated a SOI proximity effect in MoSe₂ induced by interfacing it with a high-Z element, Pb. This work extends proximity studies from graphene-based materials as receivers of the SOI field to TMDCs, establishing this class of materials as a promising platform for SOI engineering. In contrast to previous epitaxial systems, we employ an amorphous Pb overlayer, which circumvents **k**-space mismatch at the interface to enable efficient SOI transfer. Methodologically, we introduced an approach based on identifying local SOI-derived band gaps. At their edges, the spatial parts of the wavefunctions are nearly identical and thus respond equally to scalar hybridization and Rashba-type effects, effectively isolating the intrinsic SOI contribution due to the SOI field transfer into the host.

Applying this framework to the MoSe₂/Pb interface, we have found the SOI hotspots in the $\Gamma_1$- and H-points of the 3D-BZ. The Pb overlayer enhances $\Delta_{bs}$ in these points by up to ~30 meV and ~60 meV, respectively, saturating above 2 MLs. These values significantly exceed those observed in graphene-based systems and fall into the energy range relevant for spintronic applications. Owing to the intrinsic nature of the SOI hotspots, these enhancements manifest the profound intrinsic proximity effect resulting from hybridization of Mo $d$-orbitals in the top MoSe₂ layer to the Pb $sp$-orbitals, transferring a strong SOI field into the host. We have confirmed this proximity mechanism by critical assessment of all other possible effects, including chemical reactions, structure- or field-induced Rashba splitting, all of which are ruled out by experimental or theoretical

evidence. A control experiment, where changing Pb to the low-Z element Al has completely extinguished the SOI proximity effect, has verified our conclusion. Instrumental for our study has been the use of soft-X-ray ARPES, whose large probing depth was essential to penetrate through the Pb film to MoSe$_2$, and the sharp intrinsic $k_z$ definition essential to isolate the SOI hotspots in 3D **k**-space.

Our results establish a clear strategy for manipulating intrinsic SOI in TMDCs via proximity to high-Z elements. Their amorphous overlayers eliminate potential effects of **k**-space mismatch between the proximitized wavefunctions at the interface. This strategy naturally extends to TMDC intercalated with high-Z atoms. Concerning the Pb/MoSe$_2$ interface specifically, we find its device perspective attractive owing to the absence of Fermi-level pinning, consistent with weak chemical interaction and favorable Schottky contact behavior.

In a broader perspective, our demonstration of proximity-induced modulation of the intrinsic SOI is not only interesting for fundamental solid-state physics, it also illustrates a viable approach for tuning spin-related properties via heterostructure design. For example, PbS/MoSe$_2$ heterostructures could enable similar SOI control while preserving semiconducting behavior, making them suitable for integration into van der Waals transistors. The use of amorphous overlayers simplifies fabrication compared to epitaxial growth. More broadly, combining materials with distinct electronic properties (such as superconductivity and ferromagnetism) offers a route to engineer multifunctional systems for applications in spintronics, quantum technologies, and beyond.

## Methods

*ARPES measurements.* Synchrotron-radiation ARPES experiments over a series of four 2H-MoSe$_2$ bulk crystals (HQ Graphene) were performed at the SX-ARPES endstation[45] of the ADRESS beamline[67] at the Swiss Light Source (Villigen, Switzerland). The spectra were recorded in the $h\nu$ range 700–850 eV with circularly polarised light, using the hemispherical analyzer PHOBOS-225 (SPECS GmbH) at a combined (beamline + analyzer) energy resolution of ~100–120 meV. The samples were cleaved in-situ and measured at 14 K in a vacuum better than $2 \cdot 10^{-10}$ mbar.

A particular concern in acquisition of the $h\nu$-dependent ARPES data was a slight $h\nu$-dependent charging of pristine 2H-MoSe$_2$ crystals and systematic interpolation errors of the JENOPTIC monochromator energies[68]. To solve this problem, acquisition of the ARPES images for every next $h\nu$ value was followed by a short measurement of the Mo 3$d$ core-level. The energy scale in these images was then corrected by the core-level energy shift relative to its position at $h\nu = 737$ eV, where $k_z$ falls in the VBM, see Fig. 4b.

If not stated otherwise, all figures in the main text and SI represent the raw ARPES intensity, including the incoherent background coming from the Debye-Waller factor and the amorphous Pb overlayer. Reproducibility of the ARPES data measured from different 2H-MoSe$_2$ crystals is illustrated by the EDCs in the SOI hotspots shown in Fig. SI6. Conversion of the photoelectron kinetic energies and emission angles to the **k** values included the photon momentum correction[45]. The $\Delta_{bs}$ values were determined from the second derivative of the EDCs, and their confidence-interval halfwidths calculated by multiplying the standard deviation over four crystals by the Student's $t$-factor for a confidence interval of 75%[69].

*Pb deposition.* The Pb overlayers were deposited at the sample temperature 14 K. Assuming their homogeneity, the thickness ($d$) was estimated based on the attenuation of the Mo 3$d$ and Se 3$d$ core levels relative to bare MoSe$_2$ expressed as $I/I_0 = e^{-d/\lambda_{Pb}}$, where $\lambda_{Pb}$ is the IMFP in Pb. According to the TPP-2M formula[58,59], $\lambda_{Pb}$ at $h\nu = 1000$ eV equals ~19 Å for Mo 3$d$ and ~22 Å for Se 3$d$. The calibrated deposition rate was ~6 Å/min.

*DFT calculations.* Electronic band structure calculations for bulk MoSe$_2$ with and without SOI were carried out within the density functional theory (DFT) using the linear muffin-tin orbital (LMTO) method[70]. We use the fully relativistic PY LMTO computer code[71]. Self-consistent calculations have been performed with the experimental lattice parameters and within the local density approximation (LDA).

## Data availability

The raw data generated in this study, sorted in figure order, have been deposited in the database figshare.com under https://doi.org/10.6084/m9.figshare.30631352.

## Code availability

The software MATools for ARPES data processing, including **k**-space conversion, extraction of various **k**-space cross-sections, and various data visualisations, is available under https://www.psi.ch/en/sls/adress/manuals.

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

## Acknowledgements

F.A. acknowledges the financial support from the Swiss National Science Foundation within the Grant No. 200020B_188709, and J.M. from the project "Quantum materials for applications in sustainable technologies" (QM4ST) funded as project No. CZ.02.01.01/00/22_008/0004572 by Programme Johannes Amos Commenius within the call Excellent Research. The authors thank J.-H. Dil, I. Di Marko and E. Della Valle for valuable advice, and A. G. Rybkin for expert suggestions on the manuscript.

## Author contributions

V.N.S. and J.M. conceptualised the research. F.A. performed the experiment supported by V.N.S., X.W., and T.S., as well as processed the data supported by V.N.S. and P.C. J.M., D.N., and A.P. performed the DFT calculations. V.N.S., J.M., and F.A. developed the data interpretation in discussions with other authors. F.A. and V.N.S., supported by J.M., wrote the manuscript with advice from all authors.

## Competing interests

The authors declare no competing interests.
