## [Transparent Peer Review file · Nature Communications]

k-dependent modulation of intrinsic spin-orbit interaction in MoSe₂ induced by proximity to amorphous Pb

Corresponding Author: Dr Vladimir Strocov

Version 0:

Reviewer comments:

Reviewer #1

(Remarks to the Author)

Report on NCOMMS-25-04408 F. Alarab et al.

The authors present a combined experimental and theoretical study of spin-orbit effect in MoSe₂, where spin orbit interaction strength is amplified by proximity with poly-Pb layer. The experimental results are of high level, well supported by theoretical calculation;

analysis of the data is carefully performed. The work matches the high standard of Nat.Comm, yet clarification/better description of some aspects is needed before publication.

As a general comment, a more detailed discussion and/or more comments describing what is the novelty of the presented results, with a clear message to a general audience, are needed to make the work conclusive with respect to previous literature. Abstract and introduction should contain these statements.

Main comments

- 1) The second paragraph of the introduction (from "SOI is known.." to "... next-generation quantum devices") is long and connected to the main results of the present work only in a quite generic way. It makes the introduction heavy to read; it should be condensed/reshaped into less, more focused, sentences
- 2) Oppositely, in the third paragraph, the difference between extrinsic and intrinsic contributions of the proximity effect should be better explained. This point is one of the main claims of the work, yet references are missing and, most important, it is not clear how the two mechanisms could be disentangled experimentally (apart from the energy size, and from the DFT calculations in the pristine case), hence how one may assign the nature of the observed effect. Would a scheme/energy diagram help in explaining this important point?
- 3) The "transfer of SO field" linked to the Pb-MoSe₂ hybridization, which should explain the increased band splitting upon Pb deposition, seems in contradiction with the minor modifications to the band structure and with the almost absent chemical interaction (as from core level). This should be discussed to avoid confusion.
- 4) In order to directly inspect the increase of the splitting, the reader would benefit from the direct comparison (in the same figure) of both the EDCs and of second derivatives at the various thicknesses, at both H and G₁ points. This could be accomplished also in supplementary information. Also, the EDCs from some of the different measured samples should be included in the supplementary information
- 5) Fermi level of Pb is hardly visible even at high coverage in Fig. S1, as stated in pag. 9. A different image contrast should be applied or an angle integrated spectrum ? should be included to better highlight this experimental result
- 6) About the discussion of possible effect of electric field due to charge transfer (pag. 11, "Second, we have investigated..."), is this necessary? As the core level spectra and the band structure do not show any charge transfer, as explained in the previous paragraph

Minor comments

1) pag. 4, "spectral peak width increasing with EB because of the increasing hole lifetime": is it "decreasing hole lifetime"?
2) pag. 6, "(from EB -5.5 to -4.5 eV and from -3.5 to -1 eV, respectively)": between -5.5 and -4.5 eV there are no bands in fig. 4a left, the dispersive band is below -5.5 eV. 3) Pag. 10, "Another interesting fact is the slight reduction...": this reduction is totally within the error bars.

2) Pag. 11, "The behavior, is at odds with our experimental data, where no splitting of the bands... is observed upon deposition of Pb": is it "at K point" missing?

3) Fig.5

-panel a) Quantification of the integrated intensities of core levels would confirm the absence of chemical reaction;
- A photon energy of 1000 eV is not the best suited to see tiny variation in fraction of a nm thickness. Did the authors use also other (lower) photon energies?
- there is no panel d) as indicated in the caption

Reviewer #2

(Remarks to the Author)

Reviewer #3

(Remarks to the Author)

In the manuscript entitled "Strong k-dependent modulation of spin-orbit interaction in MoSe₂ induced by proximity to amorphous Pb", F. Alarab et al., presented a well-executed experimental study on the proximity-induced effects of spin-orbit interaction (SOI) in MoSe₂ by high-Z metal Pb.

The authors provide direct evidences of the modification of spin-orbit band splitting at G and K points in 2H-MoSe₂, affected by amorphous Pb films, using soft x-ray ARPES to minimize the effect of k_z broadening in ARPES measurement. They argue that the proximity to Pb significantly modulates the SOI in 2H-MoSe₂, reaching tens of meV at specific points in the Brillouin zone (BZ). The obtained proximity effect is several times stronger than similar effects observed in graphene.

Although these results are quite interesting and represent a significant contribution to a field of spintronics, there are some issues need to be clearly addressed and the electronic characterization of the investigated compound are not convincingly supported by the results presented, as detailed below. Therefore, I cannot recommend this manuscript for publication in Nature Communications.

Here are my comments:

(1) The authors display ARPES band structures aligned to binding energy of 2H-MoSe₂.

However, it looks like they were aligned to valence band maximum.

To fully eliminate the possible effects of Pb such as the charge transfer, energy of the ARPES band structure should be aligned to Fermi energy considering the semiconducting nature of 2H-MoSe₂.

(2) As described in method section, all ARPES data were taken at 14 K.

Considering an energy gap of 2H-MoSe₂ (~1.4 eV), however, there may exist the charging effect at low temperature, and this effect hinders the analysis of the spectra.

The authors must discuss the effect of charging effects in ARPES data to clearly deliver the main messages.

(3) For the effect of thickness- and coverage-dependent Pb films, authors mentioned that the SOI in 2H-MoSe₂ is enhanced with increasing Pb coverage and thickness up to 2 ML by proximity effects, arguing that the Pb coverage is evidently a key parameter to control the magnitude of Δ SO.

Then, a question is how could the proximity SOI only change up to 2 ML of Pb films as shown in Fig. 7?

Considering the Pb-overlayer is amorphously formed on 2H-MoSe₂ as the authors argued, it may be gradually proportional to the volume of Pb.

It would be great that this discussion becomes more clear.

(4) Even though authors describe that "no changes of Δ bs in either SOI hotspot were detected for Al deposition on 2H-MoSe₂" at page 11, the supplementary data show reduction of SO split at G point compared to bare 2H-MoSe₂ whereas the one at K point remains.

Authors should clearly discuss it with more detail.

The discussion at current version is not enough to fully understand and support the arguments to readers.

Version 1:

Reviewer comments:

Reviewer #1

(Remarks to the Author)

Report - revised version of NCOMMS-25-04408A by Fatima Alarab et al.

The present revised version significantly improved the focus of the paper, the quality of the discussion and the clarity of the presentation. Although we support publication in Nature Communication, we ask the authors to further clarify the following two points, as we believe such clarification is crucial to improve the quality of the presentation and the significance of their results

1) The origin of the splitting at K point is debated in literature and the outcome of the theoretical calculations conducted by the authors that such splitting could have different contributions depending on the k_z (Fig. 2) is very interesting. Since the authors have also measured the k_z dispersion (Fig. 3a), a (even tentative) direct experiment-theory comparison of the evolution of the splitting between K and H point would be highly desirable. Moreover, some references from literature should be included at the end of the sentence "It was shown that the band splitting [...] arises mostly from the interlayer interaction [...]".

2) Put aside the identification of the "SOI hotspots" through the comparison of the calculation with and without SOC, the main indication towards an intrinsic nature of the increase of the splitting lies in the observation: "the Δ_{bs} modulations in the SOI hotspots are unaffected by scalar hybridization of the Pb states which contribute to the marginal k -dependent energy shifts". Such idea is also reaffirmed many times across the paper. This claim should be better clarified, namely:

- a) Fig. 6 shows a splitting that increases upon Pb deposition, how can the effect of a scalar hybridization be excluded by inspecting these data?
- b) the ARPES comparison of bare and Pb-covered sample in Fig. 5c shows a general agreement, with some possible small shift in some points; if the authors want to use such shift as a marker of the scalar hybridization (in order to disentangle it from the intrinsic effect), they could possibly provide a more careful analysis by addressing some specific bands.

Moreover, A few typos to be corrected:

- 3) In the Introduction, "(often referred to as structure-induced Rashba effect": a closed parenthesis is missing after "Rashba effect"
- 4) In the first paragraph of Results and discussion, "This hybridization is responsible for the interlayer interaction, quantum confinement, and, for system with odd numbers of MLs, where the absence of the inversion symmetry resulted in Rashba-type SO splitting at the Gamma point": something is missing in this sentence.

This manuscript has been co-reviewed with an early career research , following the Nature Communications initiative to facilitate training in peer review and to provide appropriate recognition for Early Career Researchers who co-review manuscripts.

Reviewer #2

(Remarks to the Author)

Reviewer #3

(Remarks to the Author)

The author has carefully revised all the questions raised by the reviewers, and I recommend publishing the paper.

Reply to the Reviewers

Reviewer #1 (Remarks to the Author):

The authors present a combined experimental and theoretical study of spin-orbit effect in MoSe₂, where spin-orbit interaction strength is amplified by proximity with poly-Pb layer. The experimental results are of high level, well supported by theoretical calculation; analysis of the data is carefully performed. The work matches the high standard of Nat.Comm, yet clarification/better description of some aspects is needed before publication.

> We thank the referee for the overall positive assessment of our work, and for the numerous insightful comments. Below we answer them point by point.

As a general comment, a more detailed discussion and/or more comments describing what is the novelty of the presented results, with a clear message to a general audience, are needed to make the work conclusive with respect to previous literature. Abstract and introduction should contain these statements.

> This is an excellent point that we have somehow overlooked at the initial submission. We have added balanced statements of the experimental and methodological novelties to the abstract, introduction and outlook, which will certainly help increase the impact of our work.

Main comments

1) The second paragraph of the introduction (from " SOI is known.." to "... next-generation quantum devices") is long and connected to the main results of the present work only in a quite generic way. It makes the introduction heavy to read; it should be condensed/reshaped into less , more focused, sentences

> We acknowledge that this paragraph is quite broad in its content. However, we believe that such an overview of SOI-related phenomena is important for the general readership of Nature Comm. which includes non-specialists. Nevertheless, we have revised the paragraph to improve its focus and trim less impactful references, while preserving the context necessary for showing the broader significance of our work.

2) Oppositely, in the third paragraph, the difference between extrinsic and intrinsic contributions of the proximity effect should be better explained. This point is one of the main claims of the work, yet references are missing and, most important, it is not clear how the two mechanisms could be disentangled experimentally (apart from the energy size, and from the DFT calculations in the pristine case), hence how one may assign the nature of the observed effect. Would a scheme/energy diagram help explaining this important point?

> This is indeed a very important point that was overlooked in the first version of the manuscript. Following the reviewer's advice, we have extended the third paragraph by a basic formalism which clarifies the root distinction between the (extrinsic) Rashba-type and intrinsic SOI mechanisms. Furthermore, we have cited several theoretical papers explaining the distinction, albeit tailored to graphene-based systems. We have also elucidated that experimentally the Rashba-type mechanism can be identified based on the characteristic pattern of two band dispersions split in k -space. Otherwise, the two mechanisms can be disentangled based on DFT calculations, as demonstrated in the cited papers.

Most importantly, sharpening of the distinction between the Rashba-type and intrinsic SOI mechanisms advised by the reviewer has taken us to understanding of the point which has become of key importance

for our work. Specifically, we have realized that at the edges of the SOI band gaps the wavefunctions have almost identical spatial (spin-independent) parts and therefore respond to the scalar (non-SOI) hybridization between Pb and MoSe₂ as well as to the Rashba-type interactions in almost the same way, leaving the gap magnitude unaffected. Therefore, our measurements in the SOI hotspots directly represent the intrinsic SOI modulated by proximity to Pb. This idea has become the key methodological novelty of our work. Correspondingly, we have substantially reworked the introduction, assessment of alternative scenarios and the outlook to make them most clear to the reader. We are sincerely grateful to the reviewer for the suggestion to clarify the distinction between the Rashba-type and intrinsic SOI, which has led us to new insights of key importance.

3) The "transfer of SO field" linked to the Pb-MoS₂ hybridization, which should explain the increased band splitting upon Pb deposition, seems in contradiction with the minor modifications to the band structure and with the almost absent chemical interaction (as from core level). This should be discussed to avoid confusion.

> Discussing the experimental band structure before and after Pb deposition, Fig. 5c, we mention tangible k -dependent energy shifts whose magnitude stays at most well below 100 meV. To disentangle the proximity-induced SOI effects from these shifts, we focus on the band gaps at the SOI hotspots. In these regions, the spatial (spin-independent) parts of the wavefunctions at the gap edges are nearly identical and thus respond in the same way to non-SOI hybridization with Pb. As a result, the observed modulation of the gap width in the SOI hotspots directly reflects the true SOI proximity effect. We have slightly rearranged the text to make this point more clear.

4) In order to directly inspect the increase of the splitting, the reader would benefit from the direct comparison (in the same figure) of both the EDCs and of second derivatives at the various thicknesses, at both H and G1 points. This could be accomplished also in supplementary information. Also, the EDCs from some of the different measured samples should be included in the supplementary information

> Following this recommendation, included in Supporting Information are now the EDCs and their second derivatives for various Pb thicknesses (Fig. SI3) as well as the EDCs from four different samples (Fig. SI6). These additional datasets are referenced in the relevant sections of the manuscript.

5) Fermi level of Pb is hardly visible even at high coverage in Fig. S1, as stated in pag. 9. A different image contrast should be applied or an angle integrated spectrum ? should be included to better highlight this experimental result

> Figs. SI1-1 and SI1-2 in Supporting Information now include the angle-integrated spectra, where the Pb-derived Fermi level is clearly seen (though less clearly for the 0.25-nm coverage).

6) About the discussion of possible effect of electric field due to charge transfer (pag. 11, "Second, we have investigated..."), is this necessary? As the core level spectra and the band structure do not show any charge transfer, as explained in the previous paragraph

> We thank the referee for pointing this out. We acknowledge that invoking the charge transfer in this context was incorrect. The charge transfer (or chemical reaction) is indeed excluded based on the core-level and band structure data. Actually, we meant that the Pb layer could induce a perpendicular electric field, which might in principle modulate Δ_{bs} via a Rashba-like mechanism. This is why we performed calculations including electric fields of varying strength, which have demonstrated that such mechanisms can also be ruled out. We have revised the text to clarify the actual physics involved.

Minor comments

1) pag. 4, "spectral peak width increasing with EB because of the increasing hole lifetime": is it "decreasing hole lifetime"? 2) pag. 6, "(from EB -5.5 to -4.5 eV and from -3.5 to -1 eV, respectively)": between -5.5 and -4.5 eV there are no bands in fig. 4a left, the dispersive band is below -5.5 eV. 3) Pag. 10, "Another interesting fact is the slight reduction...": this reduction is totally within the error bars.

> The points 1) and 2) fixed. Regarding point 3), the observed slight reduction of Δ_{bs} is actually at the border of the error margins. Moreover, this reduction is seen for both Γ_{31} and Γ_{32} points, adding statistical confidence to this observation. Therefore, we decided to mention it in the text.

2) Pag. 11, "The behavior, is at odds with our experimental data, where no splitting of the bands... is observed upon deposition of Pb": is it "at K point" missing?

> The K-point was indeed missing there. Fixed.

3) Fig.5

-panel a) Quantification of the integrated intensities of core levels would confirm the absence of chemical reaction;

> We expected that any chemical reaction between MoSe₂ and Pb would change the energy or lineshape of the Mo 4d and Pb 4f core levels upon the progressive deposition of Pb. No such changes have been observed in our data. This makes the integral-intensity analysis redundant in this case.

- A photon energy of 1000 eV is not the best suited to see tiny variation in fraction of a nm thickness. Did the authors use also other (lower) photon energies?

> We are after the interface states underneath the Pb layer with a thickness of up to 2 nm. This has motivated our choice of the relatively high photon energy.

- there is no panel d) as indicated in the caption

> Fixed.

We thank the reviewer again for his insightful remarks which have not only helped us improve the scientific rigor and clarity of the manuscript, but have also elevated it to a higher level of scientific impact. We hope that the clarifications and updates provided in the revised version fully address all concerns raised and that the manuscript is now suitable for publication.

Reviewer #3 (Remarks to the Author):

In the manuscript entitled "Strong k-dependent modulation of spin-orbit interaction in MoSe₂ induced by proximity to amorphous Pb", F. Alarab et al., presented a well-executed experimental study on the proximity-induced effects of spin-orbit interaction (SOI) in MoSe₂ by high-Z metal Pb.

The authors provide direct evidences of the modification of spin-orbit band splitting at G and K points in 2H-MoSe₂, affected by amorphous Pb films, using soft x-ray ARPES to minimize the effect of k_z broadening in ARPES measurement.

They argues that the proximity to Pb significantly modulates the SOI in 2H-MoSe₂, reaching tens of meV at specific points in the Brillouin zone (BZ). The obtained proximity effect is several times stronger than similar effects observed in graphene.

Although these results are quite interesting and represent a significant contribution to a field of spintronics, there are some issues need to be clearly addressed and the electronic characterization of the investigated compound are not convincingly supported by the results presented, as detailed below. Therefore, I cannot recommend this manuscript for publication in Nature Communications.

We thank the reviewer for careful reading of our manuscript and the overall positive assessment of our work. Below, we address all of the reviewer's comments point by point.

Here are my comments:

(1) The authors display ARPES band structures aligned to binding energy of 2H-MoSe₂. However, it looks like they were aligned to valence band maximum. To fully eliminate the possible effects of Pb such as the charge transfer, energy of the ARPES band structure should be aligned to Fermi energy considering the semiconducting nature of 2H-MoSe₂.

> Indeed, we explicitly state that "... the energy scale is set relative to the global VB maximum (VBM) ...". Possible charge transfer from Pb, mentioned by the reviewer, might indeed rigidly shift the MoSe₂. In this case the energy alignment relative to the VBM allows us to ignore this shift and directly assess the SOI modulation. Anyway, the core-level measurements in Fig. 5 evidence negligible charge transfer.

(2) As described in method section, all ARPES data were taken at 14 K. Considering an energy gap of 2H-MoSe₂ (~1.4 eV), however, there may exist the charging effect at low temperature, and this effect hinders the analysis of the spectra. The authors must discuss the effect of charging effects in ARPES data to clearly deliver the main messages.

> The reviewer is absolutely right regarding the sample charging, which was a particular concern in our experiment. In Methods, we explicitly explain: "A particular concern in acquisition of the $h\nu$ -dependent ARPES data was a slight $h\nu$ -dependent charging of pristine 2H-MoSe₂ crystals ... acquisition of the ARPES images for every next $h\nu$ value was followed by a short measurement of the Mo 3d core-level, and the energy scale in these images was corrected by the core-level energy shift relative to its position at $h\nu = 737$ eV where k_z is in the VBM, see Fig. 4(b)."

(3) For the effect of thickness- and coverage-dependent Pb films, authors mentioned that the SOI in 2H-MoSe₂ is enhanced with increasing Pb coverage and thickness up to 2 ML by proximity effects, arguing that the Pb coverage is evidently a key parameter to control the magnitude of Δ SO.

Then, a question is how could the proximity SOI only change up to 2 ML of Pb films as shown in Fig. 7? Considering the Pb-overlayer is amorphyously formed on 2H-MoSe₂ as the authors argued, it may be gradually proportional to the volume of Pb.

It would be great that this discussion becomes more clear.

> The SO field originates from atomic orbitals localized near the ionic cores, making it inherently short-range (see, for example, Zollner *et al.*, Phys. Rev. B **100** (2019) 085128). Therefore, Pb atoms located away from the interface contribute little to the SO field transferred into MoSe₂. This results in fast saturation of the observed proximity effects.

(4) Even though authors describe that "no changes of Δ bs in either SOI hotspot were detected for Al deposition on 2H-MoSe₂" at page 11, the supplementary data show reduction of SO split at G point compared to bare 2H-MoSe₂ whereas the the one at K point remains. Authors should clearly discuss it with more detail.

The discussion at current version is not enough to fully understand and support the arguments to readers.

> We have fixed the typo in the graphics. Δ_{bs} in the Γ point after the Al deposition indeed stays the same within the experimental uncertainty, in contrast to the Pb deposition. We thank the reviewer for spotting this deceptive typo.

The whole body of insightful comments that we have received from the reviewers has helped us vastly improve the scientific rigor and clarity of the manuscript. Most importantly, we have sharpened the distinction between the Rashba-type and intrinsic SOI mechanisms, and outlined the main novelties of our work (amorphous overlayer circumventing the \mathbf{k} -space mismatch, extending the proximity studies to TMDC hosts, identification of the SOI hotspots where the intrinsic proximity due to the SOI field transfer from the high-Z overlayer is disentangled from scalar hybridization and Rashba-type SOI, etc.) We hope that these critical revisions to the manuscript and the above responses fully address the reviewer's concerns, and that the manuscript is now prepared for publication in Nature Communications.

Reply to the Reviewer 1

The present revised version significantly improved the focus of the paper, the quality of the discussion and the clarity of the presentation. Although we support publication in Nature Communication, we ask the authors to further clarify the following two points, as we believe such clarification is crucial to improve the quality of the presentation and the significance of their results

We thank the referee for the remaining remarks. Below, we address them point by point and explain the corresponding minor revisions .

1) The origin of the splitting at K point is debated in literature and the outcome of the theoretical calculations conducted by the authors that such splitting could have different contributions depending on the k_z (Fig. 2) is very interesting.

Since the authors have also measured the k_z dispersion (Fig. 3a), a (even tentative) direct experiment-theory comparison of the evolution of the splitting between K and H point would be highly desirable.

> Following the reviewer's suggestion, we analysed the calculated and experimental Δ_{bs} values at the K- and H-points. This analysis has yielded somewhat unexpected results, summarized in the newly introduced sentence: "However, while our calculations predict that Δ_{bs} decreases from ~290 to 200 meV when moving from the K-point to H, the experiment shows it to stay at 210 meV essentially constant within the experimental accuracy, as evidenced by the flat energy separation of the corresponding band doublet in the out-of-plane ARPES dispersions in Fig. 4(a)." We can speculate that the reduction of the interlayer-interaction gap when moving from K to H is to some extent compensated by the parallel increase of SOI, but delving into details of this behavior is beyond the focus of our work.

Moreover, some references from literature should be included at the end of the sentence "It was shown that the band splitting [...] arises mostly from the interlay interaction [...]"

> We have added the requested references, including two new ones (Refs. 70 and 71). The corresponding sentence now reads "... absence of the inversion centre [51,70,71]."

2) Put aside the identification of the "SOI hotspots" through the comparison of the calculation with and without SOC, the main indication towards an intrinsic nature of the increase of the splitting lies in the observation: "the Δ_{bs} modulations in the SOI hotspots are unaffected by scalar hybridization of the Pb states which contribute to the marginal k -dependent energy shifts". Such idea is also reaffirmed many times across the paper. This claim should be better clarified, namely:

a) Fig. 6 shows a splitting that increases upon Pb deposition, how can the effect of a scalar hybridization be excluded by inspecting these data?

> This is precisely our point. Examining the calculations without SOI in Fig. 2 (a), we find that the bands in the Γ_1 - and H-points are degenerate. The gaps open in these points only when SOI is introduced. Consequently, the electron states just above and below the SOI-induced band gaps share the same spatial (spin-independent) parts of their wavefunctions. As a result, their energy shifts under any scalar hybridization are nearly identical, making their energy difference - the central figure of our analysis - largely insensitive to such hybridization.

b) the ARPES comparison of bare and Pb-covered sample in Fig. 5c shows a general agreement, with some possible small shift in some points; if the authors want to use such shift as a marker of the scalar hybridization (in order to disentangle it from the intrinsic effect), they could possibly provide a more careful analysis by addressing some specific bands.

> The energy shifts in Fig. 5 (c) are actually combined effects of the scalar hybridization and SOI, both modulated by the proximity to Pb. In general, these two contributions cannot be directly disentangled. The only exceptions are the SOI hotspots in Γ_1 - and H-points, where we focus our analysis.

Moreover, A few typos to be corrected:

3) In the Introduction, "(often referred to as structure-induced Rashba effect)": a closed parenthesis is missing after "Rashba effect"

> Fixed.

4) In the first paragraph of Results and discussion, "This hybridization is responsible for the interlayer interaction, quantum confinement, and, for system with odd numbers of MLs, where the absence of the inversion symmetry resulted in Rashba-type SO splitting at the Gamma point": something is missing in this sentence.

> Corrected as "... in bulk MoSe₂ and quantum confinement in its heterostructures. Furthermore, in systems with an odd number of MLs lacking the inversion centre, this hybridization modulates ..."

We thank the reviewer again for their insightful questions and valuable advice. We believe that our responses and final revisions fully address their remaining concerns and that the manuscript is now ready for publication in Nature Communications.